SciPost Physics
Submission

# Self-similar solutions for fuzzy dark matter

Raquel Galazo-Garcia[1] *, Philippe Brax[2] † and Patrick Valageas[3] *

1, 2, 3 Université Paris-Saclay, CNRS, CEA, Institut de physique théorique, 91191,
Gif-sur-Yvette, France
* raquel.galazogarcia@ipht.fr, † philippe.brax@ipht.fr, * patrick.valageas@ipht.fr

December 16, 2022

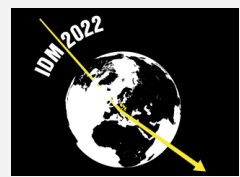

## Abstract

**Self-similar solutions for fuzzy dark matter are very different from their counterparts in the standard cold dark matter model. In contrast to the familiar hierarchical collapse of the current model for structure formation, they correspond to an inverse hierarchical blow-up. This fact highlights the gravitational cooling process, which in the absence of dissipation, allows the system to eject excess energy through the intermittent expulsion of clumps of matter. These surprising behaviours are due to the wave properties of the Schrödinger equation and to the quantum pressure. These features are printed in the Eulerian density-velocity representation of the nonrelativistic scalar field, or in the Lagrangian representation in the mass-shell trajectories.**

## 1 Introduction

Many measurements have confirmed with great accuracy that dark matter (DM) represents about 83% of the matter content of the Universe. Thanks to all this evidence, cosmologists have converged to Standard Cosmological Model, where the hypothesis of weakly interacting massive particles (> 1 Gev) [1,2] has been favoured for theoretical and experimental reasons. However, as observations and simulations have improved, a number of discrepancies between the predictions of the standard cold dark matter model (CDM) and observations on galactic and subgalactic scales have emerged. This has revived interest in alternative scenarios, including the possibility that dark matter is associated with a scalar field. One of the most attractive features of these models is the formation of soliton-like structures of large sizes that could describe the core structure of galactic halos [3]. These equilibrium configurations lead to smooth density profile [4,5] at the origin solving one of the CDM tensions at galactic scales, the core-cusp problem.

Fuzzy dark matter (FDM) [6,7] is scalar field dark matter model, $m \sim 10^{-22}$ eV that takes the simplest possibility, so the field is only subjected to gravity. The mass is extraordinarily light so that the wave nature is manifest on galactic scales providing a non-CDM behavior which leads to a different and rich phenomenology for DM on those scales. On large scales DM behaves as CDM maintaining the observable successes of CDM on those scales [3].

In this work we go beyond the static solitons by investigating dynamical self-similar solutions. With this, we access to a much deeper understanding of the dynamic processes like the observed gravitational cooling effect [8, 9]. Moreover, this allow us to obtain dynamical configurations embedded in the expanding cosmological background and to compare them with the well-known CDM self-similar solutions [10–12].

## 2 Equations of motion

The action of FDM is that of a classical scalar field $\phi$ with just a minimal coupling to gravity [7]. Variation of this action in the nonrelativistic regime, relevant for astrophysical and large-scale structures, and introducing a complex scalar field $\psi$ [6, 7], leads to the equations of motion for the complex field, known as the the Schrödinger-Poisson (SP).

We can go from the field picture to the hydrodynamical framework by taking the Madelung transformation [13] since it facilitates the comparison with the standard CDM scenario. Therefore, we can describe the system in terms of the curl-free velocity field $\vec{v}$, the density field $\rho$ and the Newtonian potential $\Phi_{\mathrm{N}}$. Now, the dynamics is encoded in the continuity and Euler equation and the Poisson equation for the gravity:

$$\frac{\partial \rho}{\partial t} + \nabla \cdot (\rho \vec{v}) = 0, \tag{1}$$

$$\frac{\partial \vec{v}}{\partial t} + (\vec{v} \cdot \nabla)\vec{v} = -\nabla \left( \Phi_{\mathrm{N}} + \Phi_{\mathrm{Q}} \right). \tag{2}$$

$$\nabla^2 \Phi_{\mathrm{N}} = 4\pi\rho. \tag{3}$$

Note that the Euler equation (2) appears an extra term, $\Phi_{\mathrm{Q}}$, which is the so called *quantum pressure* [14–16]:

$$\Phi_{\mathrm{Q}} = -\frac{\epsilon^2}{2} \frac{\nabla^2 \sqrt{\rho}}{\sqrt{\rho}}. \tag{4}$$

$\epsilon$ mimics $\hbar$ in quantum mechanics and it represents the ratio between the de Broglie wavelength $\lambda_{\mathrm{dB}}$ and the size of the system. It encodes the the wavelike effects thus in the limit $\epsilon \to 0$, we recover the usual continuity and Euler equations, which also describe CDM on large scales where shell crossing can be neglected.

## 3 Cosmological self-similar solutions

### 3.1 Self-similar ansatz

To compute self-similar solutions we look for time-dependent solutions of the following form:

$$\rho = t^{-\alpha} f\left(\frac{r}{t^\beta}\right), \quad v = t^{-\delta} g\left(\frac{r}{t^\beta}\right), \quad \Phi_{\mathrm{N}} = t^{-\mu} h\left(\frac{r}{t^\beta}\right), \tag{5}$$

By substituting these expressions into the equations (1)-(3), we find the the exponents for (5):

$$\beta = 1/2, \quad \mu = 1, \quad \alpha = 2, \quad \delta = 1/2, \tag{6}$$

Note that the functions $f$, $g$ and $h$ must be computed by solving eq. (35)-(36) of [17] .

### 3.2 Cosmological background

Since we are interested in cosmological self-similar solutions, first, we define the cosmological background using the Einstein-de Sitter universe, as for CDM. This choice is well motivate since it is a good description to the matter era when most large-scale structures are formed and because the scale factor grows as a power law of time, $a \propto t^{2/3}$, so we are not including any specific scale that would break the self-similarity. As the background expressions (see eq.(48) in [17]) are solutions of the continuity, Euler and Poisson (1)-(3) and follow the self-similar form (5) with (6) we can investigate self-similar solutions that correspond to perturbations around this expanding background.

### 3.3 Spherical self-similar solutions

We write the density and velocity fields and the gravitational potential as

$$\rho = \bar{\rho}(1 + \delta), \ \vec{v} = \bar{\vec{v}} + \vec{u}, \ \Phi_N = \bar{\Phi}_N + \varphi_N, \tag{7}$$

where $\delta$ is the density contrast and $\vec{u}$ the peculiar velocity. Using comoving coordinates $\vec{x} = \vec{r}/a$ and substituting these expressions into the continuity, Euler and Poisson equations (1)-(3), we obtain the usual comoving fluid equations (see equations (50)-(53) in [17]). Therefore, the spherical self-similar solutions are given by of the following form in agreement with (5) and (6):

$$\delta(x,t) = \hat{\delta}(\eta), \quad u(x,t) = \epsilon^{1/2} t^{-1/2} \hat{u}(\eta), \quad \varphi_N(x,t) = \epsilon t^{-1} \hat{\varphi}_N(\eta), \tag{8}$$

where we introduced the scaling variable

$$\eta = \frac{t^{1/6}x}{\epsilon^{1/2}} = \frac{r}{\sqrt{\epsilon t}}. \tag{9}$$

that includes an additional scaling in $\epsilon$ and is consistent with the self-similar exponents obtained in Sec. 3.1 .

### 3.4 Overdensities

Applying semi-analytical techniques and solving the closed equation (93) in [17] we can compute self-similar solutions for high densities around the background i.e. nonlinear regime. In this section we focus only on showing the main results for the nonlinear overdensity case of $\delta(0) = 100$.

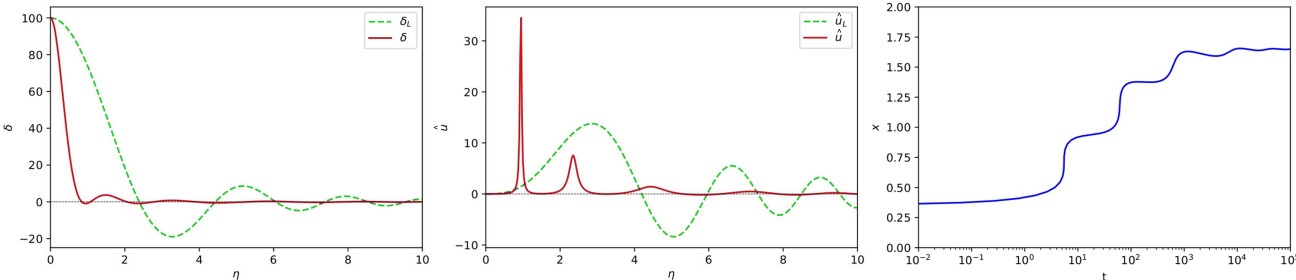

Figure 1: Overdensity case $\delta(0) = 100$. *Left panel:* nonlinear density contrast $\hat{\delta}$ (red solid line) and linear density contrast $\hat{\delta}_L$ (green dashed line). *Middle panel:* nonlinear and linear velocity fields. *Right panel:* Trajectory $x(t)$ of the comoving radius associated with a fixed mass, as a function of cosmic time $t$.

We show in red solid line in Fig.(1) the result of the nonlinear computation for the density contrast (left panel) and the velocity field (middle panel). To compare, we plot together in green dashed the result of the same overdensity but using only linear theory, i.e. solving the linear equation (73) in [17] which properly describes the small linear perturbations around the background. In this way, we can track the impact of the nonlinear corrections. We can check that they make the first peak of the density profile narrower and all higher-order peaks shift towards the center relative to the linear computation. In the velocity field, we can observe how the oscillations grow and become much sharper in contrast to the linear result. Moreover, the velocity shows high and narrow positive spikes at the density minima. Therefore, we have a scalar matter flux which is better understood in the right panel of Fig.(1) with the Lagrangian representation. This figure tracks the trajectory $x(t)$ of the comoving radius of a mass shell with mass $M$. The definition of the mass, equation (106) in [17], implicitly gives the trajectory $\eta(t)$ that we can easily rewrite in terms of the comoving coordinate $x(t)$ using (9). For the numerical computation of $x(t)$ we take $M = 1, \epsilon = 1$. Following the trajectory, starting close to the origin, inside the central peak, the comoving radius grows very slowly. Then, when the shell leaves the clump, the trajectory has an intermittent character with well-distinguishable steps because of the fast accelerations of the velocity spikes found in the velocity plot.

This ejection of matter recalls the gravitational cooling effect observed in numerical simulations. This phenomena is a mechanism that allows the system to relax towards equilibrium configurations, in spite of the absence of dissipative processes, by ejecting extra matter and energy to infinity [8,9].

## 4   High-density asymptotic limit

Since the static soliton is a huge non-linear overdensity, we can study the asymptotic behaviour of the self-similar solutions in the high-density limit. In this limit the background density becomes negligible compared to the central density peak.

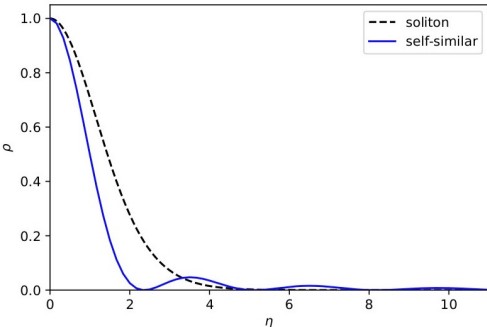

Figure 2: Asymptotic self-similar (blue solid line) and soliton (black dashed line) density profiles, normalized to $\rho(0) = 1$.

The two curves are different and the central peak of the self-similar solution is narrower than the soliton peak. This is due to the kinetic terms and because essentially the soliton balance equation is different from the self-similar Bernoulli equation (see eq.(31) and eq. (64) of [17]). Therefore, the high density asymptotic profile does not relax toward the static soliton profile. This shows that the convergence toward the soliton core is not guaranteed in all configurations.

## 5 Conclusion

CDM self-similar solutions describe the gravitational collapse of spherical overdense regions. At early times, a small linear perturbation grows, until it reaches the nonlinear regime. So transition from the linear regime to the non-linear regime occurs. However, in FDM is not the case since the amplitude of the linear perturbation remains constant in time [17]. Moreover, instead of having the familiar gravitational collapse, in FDM we have the gravitational cooling effect. In terms of the characteristic length scale (9), in FDM it grows as $\sqrt{t}$ in physical units but decreases as $t^{-1/6}$ in comoving units. Therefore, their size grows in physical units but more slowly than the scale factor, so that they actually shrink in comoving units. This feature is different from the CDM self-similar solutions that grow both in comoving size and in mass [10, 11].

## Acknowledgements

R.G.G. was supported by the CEA NUMERICS program, which has received funding from the European Union's Horizon 2020 research and innovation program under the Marie Sklodowska-Curie grant agreement No 800945. This project has received funding /support from the European Union's Horizon 2020 research and innovation programme under the Marie Sklodowska -Curie grant agreement No 860881-HIDDeN. This work was made possible by with the support of the Institut Pascal at University Paris-Saclay during the Paris-Saclay Astroparticle Symposium 2021, with the support of the P2IO Laboratory of Excellence (program "Investissements d'avenir" ANR-11-IDEX-0003-01 Paris- Saclay and ANR-10-LABX-0038), the P2I axis of the Graduate School Physics of University Paris-Saclay, as well as IJCLab, CEA, IPhT, APPEC, the IN2P3 master project UCMN and EuCAPT ANR-11-IDEX-0003-01 Paris-Saclay and ANR-10-LABX-0038.

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
