# Peer review of "Self-similar solutions for fuzzy dark matter"

_SciPost Physics Proceedings, doi:SciPost Phys. Proc. 12, 060 (2023)_

## Round 1 · Referee Report · Anonymous (Referee 1) · 2022-10-23

Strengths

  1. This manuscript studies the solutions for fuzzy dark matter halo with the additional quantum pressure term, with the self-similar ansatz.

  2. The results are interesting, in showing very different behaviour compared with the cold dark matter case. And the causes are also properly discussed.

Report

Although the study is highly mathematical, the results are interesting and useful. Thus I recommend its publication in SciPost Physics Proceedings after several minor changes.

Requested changes

  1. It will be appreciated if the author can explain a little, in the text, the physical meaning of "linear density contrast" (green line of in Figure 1) and "the linear version" (above sec.4). I guess it is directly given by Eq.(73) of [17]?

  2. For the second panel of Fig.1, should be the y-axis be labeled as "$\hat u$", as it only depends on eta?

  3. BTW, In the right panel there, is $\delta (0) = 100$ the only initial condition needed to obtain the trajectory x(t)?

  4. There are typos I suggest the author to further check, such as "that the impact of the nonlinear corrections.", and " character with well-distinguishable because".

---

## Round 2 · Referee Report · Anonymous (Referee 1) · 2022-12-18

Report

The author has addressed the concerns of the last report, and now the manuscript can be accepted.

---

## Editorial Decision

published